# Diagnostic Performance of Positron Emission Tomography with Fibroblast-Activating Protein Inhibitors in Gastric Cancer: A Systematic Review and Meta-Analysis

**DOI:** 10.3390/ijms241210136

**Published:** 2023-06-14

**Authors:** Alessio Rizzo, Manuela Racca, Federico Garrou, Elisabetta Fenocchio, Luca Pellegrino, Domenico Albano, Francesco Dondi, Francesco Bertagna, Salvatore Annunziata, Giorgio Treglia

**Affiliations:** 1Department of Nuclear Medicine, Candiolo Cancer Institute, FPO-IRCCS, 10060 Turin, Italy; alessio.rizzo@ircc.it (A.R.); manuela.racca@ircc.it (M.R.); 2Nuclear Medicine Unit, Department of Medical Sciences, AOU Città della Salute e della Scienza, University of Turin, 10126 Turin, Italy; federico.garrou@unito.it; 3Department of Oncology, Candiolo Cancer Institute, FPO-IRCCS, 10060 Turin, Italy; elisabetta.fenocchio@ircc.it; 4Department of Surgery, Candiolo Cancer Institute, FPO-IRCCS, 10060 Turin, Italy; luca.pellegrino@ircc.it; 5Division of Nuclear Medicine, Università degli Studi di Brescia and ASST Spedali Civili di Brescia, 25123 Brescia, Italy; domenico.albano@unibs.it (D.A.); f.dondi@outlook.com (F.D.); francesco.bertagna@unibs.it (F.B.); 6Unità di Medicina Nucleare, GSTeP Radiopharmacy—TracerGLab, Dipartimento di Diagnostica per Immagini, Radioterapia Oncologica ed Ematologia, Fondazione Policlinico Universitario A. Gemelli, IRCCS, 00168 Rome, Italy; salvatore.annunziata@policlinicogemelli.it; 7Clinic of Nuclear Medicine, Imaging Institute of Southern Switzerland, Ente Ospedaliero Cantonale, 6501 Bellinzona, Switzerland; 8Faculty of Biology and Medicine, University of Lausanne, 1011 Lausanne, Switzerland; 9Faculty of Biomedical Sciences, Università della Svizzera Italiana, 6900 Lugano, Switzerland

**Keywords:** FAPi, fibroblast-activating protein inhibitor, PET, positron emission tomography, gastric cancer, stomach, oncology

## Abstract

Various papers have introduced the use of positron emission tomography (PET) with [^68^Ga]Ga-radiolabeled fibroblast-activation protein inhibitor (FAPi) radiopharmaceuticals in different subtypes of gastric cancer (GC). Our aim was to assess the diagnostic performance of this novel molecular imaging technique in GC with a systematic review and meta-analysis. A straightforward literature search of papers concerning the diagnostic performance of FAP-targeted PET imaging was performed. Original articles evaluating this novel molecular imaging examination in both newly diagnosed GC patients and GC patients with disease relapse were included. The systematic review included nine original studies, and eight of them were also eligible for meta-analysis. The quantitative synthesis provided pooled detection rates of 95% and 97% for the assessment of primary tumor and distant metastases, respectively, and a pooled sensitivity and specificity of 74% and 89%, respectively, for regional lymph node metastases. Significant statistical heterogeneity among the included studies was found only in the analysis of the primary tumor detection rate (I^2^ = 64%). Conclusions: Beyond the limitations of this systematic review and meta-analysis (i.e., all the included studies were conducted in Asia, and using [^18^F]FDG PET/CT as a comparator of the index test), the quantitative data provided demonstrate the promising diagnostic performance of FAP-targeted PET imaging in GC. Nevertheless, more prospective multicentric studies are needed to confirm the excellent performances of FAP-targeted PET in this cluster of patients.

## 1. Introduction

With the third-highest mortality rate and the fifth-highest incidence among all solid tumors, gastric cancer (GC) poses a significant worldwide health burden [1,2]. Concerning sex and regional variation, GC can vary among different populations: men are more susceptible than women by a factor of from two to three; moreover, the frequency shows a wide geographic range, as it has been observed that developing nations account for more than half of all new diagnoses. The likelihood of GC development is higher in Central and South America, Eastern Europe, and East Asia (China and Japan), whereas North America, Australia, New Zealand, Southern Asia, and North and East Africa are among the low-risk areas [2].

The Lauren classification is the most widely used GC categorization; this classification distinguishes two kinds of GC: intestinal and diffuse [3]. Each type exhibits different features concerning clinical traits, genetics, anatomy, epidemiology, and growth properties [4]. According to this classification, intestinal GC is further divided into the tubular and glandular subtypes, which might show different levels of dedifferentiation. The diffuse variant is distinguished by weakly cohesive cells that lack glandular development and includes GC with signet ring cells, which is currently classified as a poorly cohesive form of GC, with tumor cells characterized by prominent cytoplasmic mucin and an eccentrically positioned crescent-shaped nucleus [5].

Several variables, including family history, nutrition, alcohol intake, smoking, and Helicobacter pylori and Epstein–Barr virus (EBV) infections, are identified as having a substantial effect on the higher chance of developing GC [4].

Instrumental staging provides crucial information regarding the tumor burden and is essential for developing an effective treatment plan. The use of endoscopic ultrasound (EUS), computed tomography (CT), 2-[^18^F]fluorodeoxyglucose ([^18^F]FDG) positron emission tomography/computed tomography (PET/CT), and laparoscopy have significantly enhanced the initial clinical staging of GC [6]. EUS is recommended to evaluate the tumor invasion profundity and lymph node involvement [7]. Nevertheless, the diagnostic accuracy of EUS may vary according to the operator’s expertise, and the evaluation of distant lymph nodes is also suboptimal due to the traducer’s limited depth and visibility [7]. Currently, CT is routinely used for preoperative staging and has an overall accuracy range of 43–82% for measuring the depth of invasion [6]. Concerning the use of [^18^F]FDG PET/CT in GC, the variable and occasionally intense physiological [^18^F]FDG absorption within the stomach wall can make it difficult to detect primary gastric cancers; furthermore, due to its limited spatial resolution, the T-stage cannot be reliably assessed [8]. In terms of nodal staging, [^18^F]FDG PET/CT is less sensitive than EUS; furthermore, [^18^F]FDG is unreliable for evaluating the extent of signet ring GC, as these tumors have a lower [^18^F]FDG uptake than other histological subtypes [8].

In recent years, most of the literature has enhanced a crucial concept: cancer is not restricted to malignant tumor cells alone, as it is characterized by a fundamental imbalance of the entire cell environment (TME), which is a complex dynamic system made up of cellular and non-cellular components [5]. Cancer-associated fibroblasts (CAFs) are stromal cells found lacking in epithelial, endothelial, or leukocyte markers, and, more importantly, oncogene mutations [9]. According to the literature, CAFs express as-smooth muscle actin (a-SMA) and fibroblast-activating protein (FAP). As an essential component of the TME, CAFs are directly related to tumor growth and invasion, the development of metastases, and the response to therapy [10]. To support this postulation, recent studies have reported that a high expression of CAFs was closely associated with pathological indicators related to advanced GC, including stage and lymph node and distant metastases [11]. Furthermore, recent studies have confirmed that CAFs can promote GC invasion and metastasis by inducing epithelial–mesenchymal transition, extracellular matrix remodeling, and tumor angiogenesis [11]. These effects reduce the adhesion between tumor cells and increase their motility, thereby facilitating their separation from the primary site and spread to other locations.

As already stated, in the tumor stroma, CAFs can express FAP, an atypical type II transmembrane serine protease, at high levels. FAP is usually undetectable in normal adult tissues, whereas its expression is significantly elevated in sites of tissue remodeling, such as inflammation and tumors [12,13]. According to these findings, FAP has become a potential target for the molecular imaging of many tumors, as well as non-oncological diseases, and FAP-targeting radiopharmaceuticals based on FAP-specific inhibitors (FAPi), including [^68^Ga]Ga-FAPi-02 and [^68^Ga]Ga-FAPi-04, have been developed [13].

Several recent studies employed PET imaging with radiolabeled FAPis to detect GC lesions in different clinical settings. The purpose of this paper is to conduct a systematic review and meta-analysis to determine the diagnostic performance of PET with radiolabeled FAPi in GC patients. This article’s secondary purpose is to collect evidence comparing the diagnostic performance of FAPi PET and other imaging techniques in GC patients.

## 2. Materials and Methods

### 2.1. Protocol

The current systematic review and meta-analysis was carried out in accordance with a preset protocol, and the “Preferred Reporting Items for a Systematic Review and Meta-Analysis” (PRISMA 2020 statement) served as a guideline for its development. The complete PRISMA checklist is available in the Appendix A. No prior registration was conducted. As a first stage, a straightforward review query was formulated: What is the diagnostic performance of FAPi PET in gastric cancer? In accordance with the Population, Intervention, Comparator, and Outcomes (PICO) framework, a literature search was conducted to establish the following criteria for study inclusion: patients diagnosed with GC (Population), undergoing PET with an FAPi compared or not with standard-of-care imaging (Comparator); the assessment of the FAPi uptake in GC and the FAPi PET DR in GC patients were defined as the outcomes of interest. Three investigators (A.R., G.T., and F.G.) independently conducted the literature search, study selection, and quality evaluation. A consensus meeting resolved all disagreements between reviewers.

### 2.2. Strategy for Literature Research and Information Sources

After defining the review question, a comprehensive literature search was conducted using two electronic scholarly databases (PubMed/MEDLINE and Cochrane Library) to identify publications evaluating the diagnostic accuracy of FAPi PET in GC patients. The ClinicalTrials.gov database was additionally searched for ongoing investigations (access date: 12 April 2023). A search algorithm based on the following terms was employed: (A) “PET” OR “positron” AND (B) “FAPi” AND (C) “gastric” OR “stomach”. There were no restrictions regarding the year of release or article language. Furthermore, the references from included studies were scrutinized for additional articles that could be used to strengthen the research. The last update to the literature inquiry was put in place on 12 April 2023.

### 2.3. Eligibility Criteria

The authors considered eligible for inclusion in this meta-analysis and systematic review clinical investigations reporting data concerning the use of FAP-targeted PET in the staging and restaging of GC. Studies including tumor types other than GC in their analyses, reviews, letters, remarks, editorials on the topic of interest, case reports or small case series on the analyzed subject, and original articles from other disciplines were excluded from the analysis. With regard to the meta-analysis (quantitative analysis), studies were excluded if there was a lack of sufficient information for pooled analyses or the potential overlap of patient data with another study.

### 2.4. Selection Method

The titles and abstracts of the obtained papers were evaluated in accordance with the preconceived eligibility criteria. The final decision concerning the inclusion of the selected studies was conducted independently for the systematic review and meta-analysis.

### 2.5. Process of Data Collection and Data Extraction

To avoid potential biases, the researchers separately gathered each of the studies included and extracted data through the information in the entire manuscript, tables, and images. For each study included in the systematic review, the following data were extracted: overall study information (authors, nation, release year, methodology, and financing sources); patient details (sample dimension, gender, age, clinical setting, and additional instrumental examinations); index test details (administered radiopharmaceutical, kind of hybrid imaging procedure, patient preparation, administered activity, and uptake time between radiolabeled FAPi administration and image acquisition).

### 2.6. Quality Assessment (Risk of Bias Assessment)

QUADAS-2, a tool for evaluating the quality of studies on the accuracy of diagnostic procedures, was used to analyze the risk of bias in individual studies and their relevance to the review query. The authors evaluated the quality of the included studies in the systematic review and meta-analysis independently. Four domains (patient selection, index test, reference standard, and flow and timing) were evaluated for bias risk, while three sectors were assessed for applicability (patient selection, index test, and reference standard).

### 2.7. Effects Metrics

The detection rate of FAPi PET in primary tumor and distant metastases and the sensitivity and specificity in metastatic lymph node assessment were the main outcomes of the meta-analysis. The secondary outcome measures were characterized in the qualitative synthesis (systematic review), taking into consideration the information presented in the results sections of the included studies.

### 2.8. Statistical Analysis

All diagnostic accuracy measures were calculated on a per-patient-based analysis. As suggested by DerSimonian and Laird, the authors performed a pooled analysis concerning the detection rate of FAP-targeted PET in primary tumor and distant metastases and a pooled analysis of the sensitivity and specificity for the detection of nodal metastases using data from the included studies, accounting for the weight of each study using a random-effects statistical model. In addition, 95% confidence interval values were provided and then illustrated using forest plots. The I-square index or inconsistency index was used to estimate statistical heterogeneity among the included studies; statistical heterogeneity was deemed significant if the I-square index was greater than 50%. In addition, publication bias was evaluated via a visual examination of the symmetry/asymmetry of the funnel plot, or by employing Egger’s test to determine whether fewer than six studies were included in the meta-analysis. Detection rate calculations were conducted using MedCalc^®^ statistical software (v. 18.2.1, bvba, Ostend, Belgium), whereas OpenMeta[Analyst]^®^, a software funded by the Agency for Healthcare Research and Quality (AHRQ) (v. 12.11.14, Rockville, MD, USA), was used for the calculation of the pooled sensitivity and specificity.

### 2.9. Additional Analyses

In the event of statistically significant heterogeneity among the included studies, subgroup analyses based on study design, patient characteristics, technical aspects, and investigated clinical settings were conducted.

## 3. Results and Discussion

### 3.1. Literature Search and Study Selection

Overall, 48 records were found in the comprehensive literature search (last update: 12 April 2023). As stated in the Materials and Methods section, these 48 publications were evaluated for eligibility based on the predetermined inclusion and exclusion criteria, and 39 records were rejected (14 not on the topic of interest; 14 as case reports; 5 as reviews; 1 as an original article retracted from publication; 1 as a retraction note; 4 as original papers also enrolling patients with tumors other than GC). After full-text evaluation, the nine remaining studies were assessed as eligible for inclusion in the systematic review (qualitative synthesis) [14,15,16,17,18,19,20,21,22]. All the studies included in the qualitative synthesis but one were classified as eligible for the subsequent meta-analysis (quantitative synthesis) [14,15,17,18,19,20,21,22]. After reviewing the references for these articles, no other studies meeting the inclusion criteria were found. The selection of the studies is summarized in Figure 1. All the excluded studies are listed in the Appendix A.

### 3.2. Study Characteristics

Table 1, Table 2 and Table 3 give the thorough feature analysis of the nine studies that met the inclusion criteria for the systematic review (qualitative analysis), which included a total of 280 GC patients. According to the general study information (Table 1), the included studies were published from 2021 to 2023 in China (7/9), Turkey (1/9), and Israel (1/9). Five of the included studies accounted for a prospective design [15,17,18,19,20], while the remaining four were retrospective [14,16,21,22]. Moreover, seven studies were conducted in a single center [15,16,17,18,19,20,21], whereas the reamaining two were bicentric [14] and multicentric [22]. Seven out of nine studies disclosed financing resources in the text [14,16,18,19,20,21,22].

With regard to the patient key characteristics (Table 2), the number of enrolled GC patients ranged from 13 to 62 (mean/median age range: 51–70 years; male percentages ranged from 46% to 71%). The index test was employed only for staging in two papers [14,19], and just for restaging in one article [16], and for both staging and restaging purposes in the remaining seven studies [15,17,18,20,21,22]. With regard to the histologic subtypes of GC included in every study, all the studies but one enrolled both patients diagnosed with gastric adenocarcinoma and gastric signet ring cell carcinoma, whereas the remaining one did not specify the histopathological subtype of the enrolled patients [16]. Finally, the comparative imaging examination was [^18^F]FDG PET/CT in all the studies [14,15,16,17,18,19,20,21,22].

As reported in Table 3, the main characteristics of the index test varied significantly between the included reports. In all studies, the radiopharmaceutical administered was [^68^Ga]Ga-DOTA-FAPi-04, with an activity ranging between 111 and 194 MBq when measured as absolute values, and between 1.8 and 2.2 MBq/Kg when measured as relative values [14,15,16,17,18,19,20,21,22]. Furthermore, the time between the FAP-targeting tracer injection and PET imaging ranged from 30 to 71 min. In six of the included investigations, PET images were coregistered with low-dose CT [15,16,17,18,19,21], in two they were fused both with CT and magnetic resonance (MR) [14,22], and in the remaining one, the only hybrid imaging technique used was PET/MR [20]. All the included studies performed both qualitative and semiquantitative analyses while interpreting PET images. Semiquantitative analyses were accomplished by calculating the maximal standardized uptake values (SUV_max_) and target-to-background uptake ratio (TBR) of the analyzed lesions.

### 3.3. Risk of Bias and Applicability

The overall “risk of bias and concerns” evaluation of the applicability for studies included in the systematic review according to QUADAS-2 is presented in Figure 2.

### 3.4. Results of Individual Studies (Qualitative Synthesis)

In all the investigations contained in this systematic review, based on both per-patient and per-lesion analyses and in diversified clinical scenarios, FAP-targeting PET/CT or PET/MRI demonstrated excellent diagnostic performance in detecting primary or locally recurrent GC lesions, as well as metastatic GC lesions in the lymph nodes, bones, peritoneum, ovaries, and lungs [14,15,16,17,18,19,20,21,22]. Concerning [^68^Ga]Ga-DOTA-FAPi-04 safety, its administration was well tolerated, and, when reported, none of the studies included the recording of any adverse events [15,16,21].

As stated in Table 4, which reports the SUV_max_ centrality measures and synthesizes the main results of each study, all the papers included in this systematic review but one reported the variable uptake of FAP-targeting radiopharmaceuticals in GC primary or locally recurrent lesions, as well as in lymph nodes and distant metastases; in all cases, it was higher than the surrounding physiological activity [14,15,17,18,19,20,21,22]. The mean/median values of the SUV_max_ ranged from 5.5 to 18.81 for primary tumor and local recurrence, whereas they ranged from 4.3 to 9.2 and from 4.2 to 8.0 for lymph nodes and distant metastases, respectively.

Compared to [^18^F]FDG as a PET radiopharmaceutical, radiolabeled FAPis showed a greater number of positive patients and lesions; furthermore, when the [^18^F]FDG and [^68^Ga]Ga-FAPi-04 uptake were compared in positive lesions, FAP-targeted PET/CT showed overall higher values of the SUV_max_ and TBR, even though the difference in the uptake of these two tracers was not statistically significant in some of the included studies [14,15,17,18,19,20,21,22].

Three of the included papers performed immunohistochemistry evaluations of the FAP expression on CAFs, and compared its staining on histopathologic samples with the FAP-targeting radiopharmaceutical uptake on PET images [14,16,18]. As a result, all three studies exploring this feature agree that radiolabeled FAPi uptake positively correlates with FAP expression on histopathological samples. Moreover, in one study, an increased [^68^Ga]Ga-FAPi-04 uptake was positively related to the presence of myeloid-derived suppressor cells, macrohage M2, and PD-1 immunoreactivity in neoplasms [16]. This statement induced the authors to postulate that [^68^Ga]Ga-FAPi-04 uptake might be a useful marker to predict the PD-1-targeting immunotherapy efficacy in GC patients.

### 3.5. Meta-Analysis (Quantitative Synthesis)

As stated in the Materials and Methods section, the meta-analysis was divided into three subanalyses exploring the detection rate of FAPi-targeted PET/CT in primary tumors (per-patient-based analysis), its sensitivity and specificity in assessing local lymph node involvement (both per-patient- and per-lesion-based analyses), and, finally, to assess its detection rate of distant metastases (per-patient-based analysis).

#### 3.5.1. Detection Rate of Primary Tumors

Eight studies including 225 GC patients were selected for the pooled analysis of the DR of primary tumors on FAPi-targeted PET images. Overall, the DR of PET/CT or PET/MRI with FAPi-targeted PET for detecting primary GC ranged from 90.3% to 100% [14,15,17,18,19,20,21,22] (Table 5).

The pooled DR of primary GC was 95.3 (95% confidence interval (95% CI): 91.68–97.60) (Figure 3). A moderate statistical heterogeneity among the included studies was found, as the I^2^ was 68%. Finally, a funnel plot for publication bias assessment (Figure 3) showed no significant asymmetry, supporting the absence of significant publication biases.

Based on the reported statistical heterogeneity, a subgroup analysis omitting the only study that enrolled only patients with signet ring cell GC [22] was performed. The subgroup analysis showed a pooled DR of 96.72% (95% confidence interval: 93.32–98.68) without significant statistical heterogeneity among the included studies (I^2^: 40.62%).

#### 3.5.2. Sensitivity and Specificity in Lymph Node Metastases

Eight studies reporting the diagnostic accuracy of FAPi-targeted PET in lymph node assessment in 147 GC patients were included in this subgroup analysis.

Based on a per-patient analysis, the pooled sensitivity and specificity of PET (coregistered with CT or MR) with a radiolabeled FAPi in the assessment of local lymph node metastases were 0.75 (range: 0.58–0.87) and 0.89 (range: 0.76–0.95), respectively. A summary receiver operating characteristic (SROC) curve is reported in Figure 4, and a forest plot is shown in Figure 5.

The pooled positive and negative likelihood ratios and the diagnostic odds ratio were 4.38 (95% CI: 1.82–10.529), 0.16 (95% CI: 0.09–0.28), and 25.68 (95% CI: 8.25–79.91), respectively (Figure 6 and Figure 7). There was no significant statistical heterogeneity among the studies included in this subanalysis, as the inconsistency index was always below 50%.

Because only four studies reported complete per-lesion analyses, a meta-analysis was not feasible in this context [15,18,21,22].

#### 3.5.3. Detection Rate of Distant Metastases

Eight studies evaluating the presence of distant metastases in 82 GC patients were selected for the pooled analysis of the DR of distant metastases on FAPi-targeted PET images. Overall, the DR of PET/CT or PET/MRI with FAPi-targeted PET for detecting GC distant metastases ranged from 91.67% to 100% [14,15,17,18,19,20,21,22] (Table 6).

The pooled DR of GC distant metastases was 96.90 (95% confidence interval: 90.90–99.41) (Figure 8). As the inconsistency index was 0%, no significant statistical heterogeneity among the included studies was found; moreover, the funnel plot for publication bias assessment (Figure 8) did not enhance significant asymmetry, supporting the absence of significant publication biases.

### 3.6. Discussion

Due to its overexpression on the cell surface of stromal cells in the tumor microenvironment, FAP is a novel potential target for molecular imaging and, perhaps, radioligand therapy [13]. In recent years, the literature concerning the employment of FAP-targeted PET imaging in oncology has been gradually growing, giving a hint as to what its applications could be. Recent studies report that FAP-targeted PET accomplished good performances in the diagnostics of different tumors, including neoplasms usually characterized by low or absent [^18^F]FDG avidity [23,24]. Furthermore, the muscle and blood-pool background of FAP-targeting radiopharmaceuticals on PET images is usually very low, resulting in a higher TBR and, subsequently, a superior image quality to [^18^F]FDG PET imaging [25].

In the past three years, several papers have tried to assess the diagnostic performance of PET imaging with FAP-targeting radiopharmaceuticals to detect GC lesions in newly diagnosed patients, as well as in patients who previously underwent surgery or chemotherapy [14,15,16,17,18,19,20,21,22]. This meta-analysis pooled the currently available data to increase the statistical power and accomplish a more robust estimate of FAP-targeted PET performances than the single original studies.

Concerning the overall diagnostic performance of PET imaging with FAP-targeting radiopharmaceuticals in GC patients, excellent accuracy was recorded both in the initial staging as well as in the restaging setting. All the studies in the meta-analysis compared FAP-targeted PET (coregistered with CT or MR) to [^18^F]FDG PET imaging [14,15,17,18,19,20,21,22].

With regard to the detection of primary GC, all the studies included in this systematic review and meta-analysis reported an overwhelming superiority of FAP-targeted PET over [^18^F]FDG PET imaging, reaching a pooled detection rate slightly over 95%. Among the included studies, the diagnostic impact of radiolabeled FAPis was even superior in patients diagnosed with signet ring cell carcinoma, which often shows weak or absent [^18^F]FDG uptake due to its lack of expression of the glucose transporter 1 (GLUT-1) transporter [14,15,17,18,19,20,21,22]. In one of the included studies, Jiang et al. observed that FAP-targeted PET/CT has a detection rate of 100%, even in small-sized tumors with the longest diameter inferior or equal to 4 cm [14]. Furthermore, some studies have reported that the depth of invasion of primary GC, a crucial prognostic factor and one of the most important features guiding the patient’s management, might affect the uptake values, with higher SUV_max_ in invasive GC [14,18,19,20,21].

As for primary tumor assessment, [^68^Ga]Ga-FAPi-04 PET showed a slightly superior diagnostic performance compared to [^18^F]FDG PET imaging both in the sensitivity and specificity of regional lymph node metastases, with a pooled sensitivity and specificity of 75% and 89%, respectively, on a per-patient-based analysis [14,15,17,18,19,20,21,22]. When compared to the diagnostic accuracy of primary tumors, the performances of FAPi-targeted PET seem to be underpowered; this statement might be explained by two factors: first, the reported size of regional lymph nodes might be smaller than the spatial resolution of PET scanners, and, in this context, the employment of fluorine-labeled FAPi tracers, characterized by lower positron energy, might improve the lymph node metastases assessment; second, the uptake of small perigastric lymph nodes might be covered by the radioactive volume effect of the primary GC due to stomach motility respiratory movements.

Although histopathological analysis is currently the gold standard for the assessment of distant metastases, non-invasive imaging has grown as a milestone in the staging of oncologic patients, and GC is no exception. In the included studies, FAP-targeted PET imaging outperformed [^18^F]FDG PET, detecting higher numbers of distant metastases both in typical and atypical sites, including the peritoneum, supra-diaphragmatic lymph nodes, lungs, bone, liver, adrenal glands, and ovaries, reaching a pooled detection rate of almost 97% on a per-patient-based analysis [14,15,17,18,19,20,21,22]. Among the typical secondary sites usually involved in GC, peritoneal metastases deserve special mention, as the peritoneum is one of the most common sites of metastases, and their extent can determine whether the patient is suitable for surgery. In all the eight studies included in this meta-analysis, an excellent detection rate, much greater than [^18^F]FDG PET, was observed for the assessment of peritoneal involvement. This outstanding result may be explained by the presence of the fibrotic reaction of tumor cells invading the peritoneum. Nevertheless, the included articles state that [^68^Ga]Ga-FAPi-04 PET might not be the best technique to assess uterine and ovarian metastases, as both are sites of high physiological uptake, which can make the interpretation ambiguous.

The promising performances reported for radiolabeled FAPis in primary GC of variable dedifferentiation levels, alongside the well-known hindrances of [^18^F]FDG in the diagnostics of several GC subtypes, including signet ring cell GC, hasten the postulation of employing FAP-targeting radiopharmaceuticals as a first-choice tracer for GC diagnostics. Nevertheless, a new diagnostic technique might change patient management once its employment can up- or downstage a pathology when compared to conventional imaging and, subsequently, affect the previously planned treatment. Among the studies included, three reported an upstaging in 3/34, 5/25, and 4/62 patients, respectively [19,21,22]; these data may have been brought about by a patient selection bias, as many of the enrolled patients had metastases and therefore were not suitable for surgery. In this context, more studies are needed to clearly assess which clusters of patients may benefit from this novel molecular imaging examination.

Inflammation and fibrosis in the tumor stroma may increase FAP-targeting radiopharmaceutical uptake [26]. As often observed on [^18^F]FDG PET images, false-positive findings brought about by surgery or fibrous reaction associated with radiation therapy should be carefully assessed [27]. Indeed, in one of the included studies, an increased [^68^Ga]Ga-FAPi-04 uptake was observed next to the suture material in the duodenal stump six months after surgery, and the surrounding reactive lymph nodes were classified as false positive [17]. In this context, multicentric studies with larger sample sizes are needed to assess what the actual limitations are of this novel molecular imaging examination.

Although the relatively recent development of immune checkpoint blockade (ICB) therapies have revolutionized the management of various types of cancer, the clinical trials KEYNOTE-012 and KEYNOTE-059 reported objective response rates to pembrolizumab therapy of 22% in PD-L1-positive metastatic GC patients [28] and 11.6% in metastatic GC patients, regardless of their PD-L1 expression status [29]. Consequently, efficient biomarkers to predict the responses and prognoses of metastatic GC patients with immunotherapy are needed. In this setting, Rong et al. reported that high FAP expression in the TME might predict poor prognoses in GC patients undergoing ICB therapy and is positively correlated with immunosuppressive cell infiltration [16]. On these bases, more studies on metastatic GC patients are needed to improve the predictive value of this new molecular imaging technique, as well as to properly define its role in this context.

Based on the current literature data, more prospective multicenter studies with larger sample sizes on the diagnostic accuracy of PET/CT or PET/MRI with FAP-targeting radiopharmaceuticals in GC are required. Moreover, studies evaluating the impact of the index test on the management of GC and cost-effectiveness analyses (comparing a diagnostic approach with or without the index test) are needed to better define the role of PET imaging with FAP-targeting radiopharmaceuticals in GC.

3-deoxy-3-[^18^F]fluorothymidine ([^18^F]FLT) is a tracer that allows for studying cell proliferation. It is phosphorylated by the cytosolic enzyme thymidine kinase 1 (TK-1) and is trapped in the cell; its uptake is positively correlated with cell growth and TK-1 activity [30] and has been experimented on to study GC patients [31,32,33]. Although the current literature data do not provide studies directly comparing FAP-targeting radiopharmaceuticals to [^18^F]FLT PET imaging, the current meta-analysis reports a superior diagnostic accuracy of FAP-targeted PET both in primary tumor and nodal metastases assessment.

To date, even if several evidence-based data on FAPi PET are already available [34,35], there is only one previously published systematic review and meta-analysis that explored the potential role of FAP-targeted PET imaging in GC patients, but it was focused on the comparison between [^68^Ga]Ga-FAPi-04 and [^18^F]FDG only [36]. The present systematic review and meta-analysis gathers more articles, both in the qualitative synthesis and in the quantitative analysis; moreover, in this updated evidence-based article, a deeper analysis of lymph node metastases assessment was performed, as we accomplished a bivariate meta-analysis including pooled sensitivity and specificity.

This systematic review and meta-analysis accounts for some limitations. First, all studies included in this meta-analysis were conducted in Asia [14,15,16,17,18,19,20,21,22]; hence, there was a lack of studies from other continents. Second, in all the studies, the FAP-targeted PET diagnostic performances were compared to [^18^F]FDG PET imaging [14,15,16,17,18,19,20,21,22]; because the latter is affected by several limitations in the evaluation of some subtypes of GC, this may be a source of bias.

The heterogeneity among the studies included in a meta-analysis might be a potential source of bias [37]. Indeed, we reported significant heterogeneity among the included studies when assessing the detection rate of primary GC, as the inconsistency index was 68%. A subsequent subgroup analysis omitting the only study enrolling only signet ring cell GC patients [22] was performed and the analysis did not show statistical heterogeneity.

## 4. Conclusions

The qualitative and quantitative data provided by this systematic review and meta-analysis highlight the promising performances of FAP-targeted PET imaging for GC primary tumor, lymph node, and distant metastases detection. Nevertheless, more multicentric trials are needed to confirm these findings to precisely define the indications and timing of PET imaging with FAP-targeting radiopharmaceuticals in GC patients (especially when compared to current reference imaging examinations), and to establish specific clinical recommendations.

## Figures and Tables

**Figure 1 ijms-24-10136-f001:**
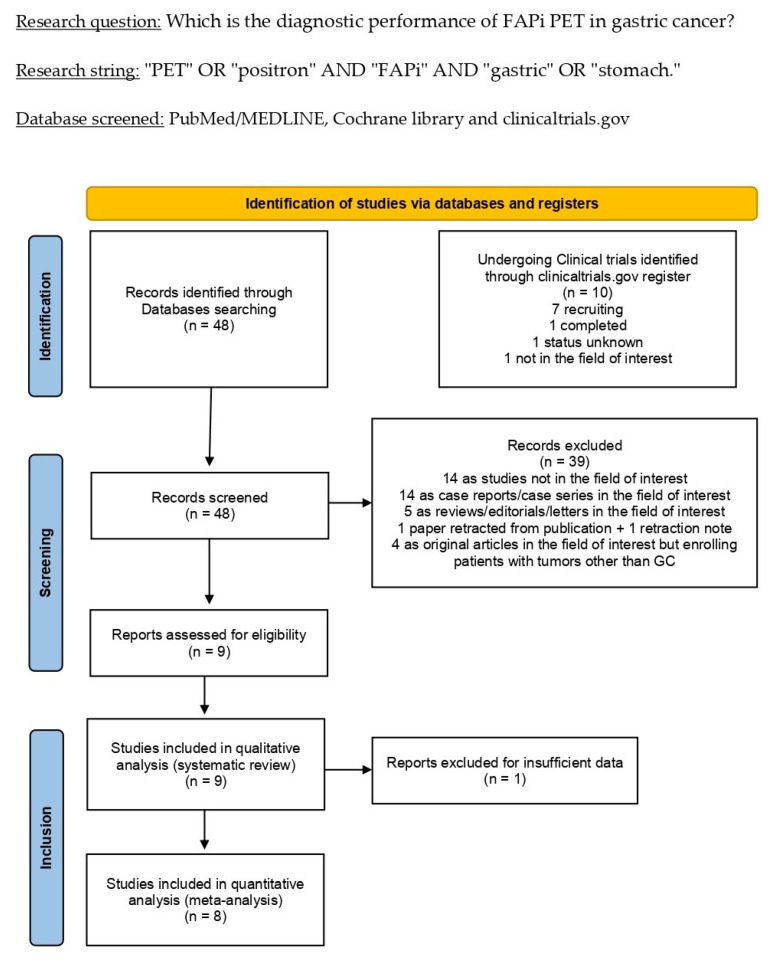
Summary of the study selection process for the systematic review and meta-analysis.

**Figure 2 ijms-24-10136-f002:**
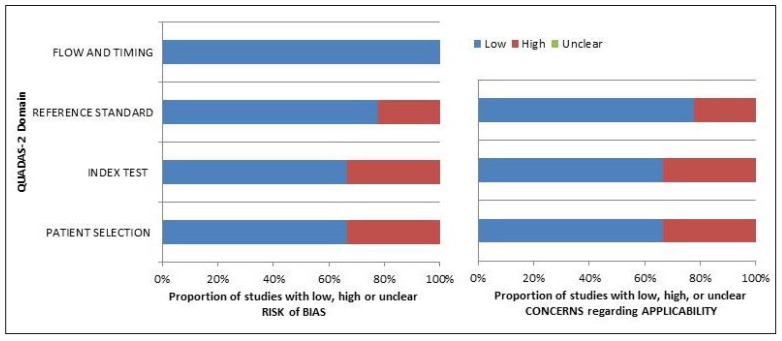
Summary of quality assessment according to QUADAS-2 tool. Studies included in the systematic review are classified as at low-risk or high-risk of bias or applicability concerns for different domains (reported in the vertical axis). The horizontal axis indicates the percentage of studies. The graph indicates that a percentage above 60% was reached in all domains.

**Figure 3 ijms-24-10136-f003:**
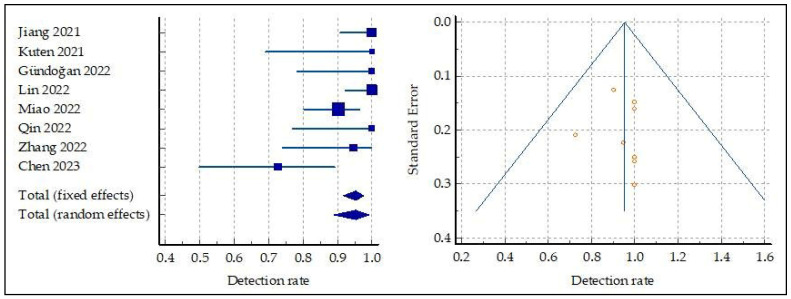
Meta-analysis and funnel plot concerning the detection rate of FAP-targeted PET in primary GC, [14,15,17,18,19,20,21,22].

**Figure 4 ijms-24-10136-f004:**
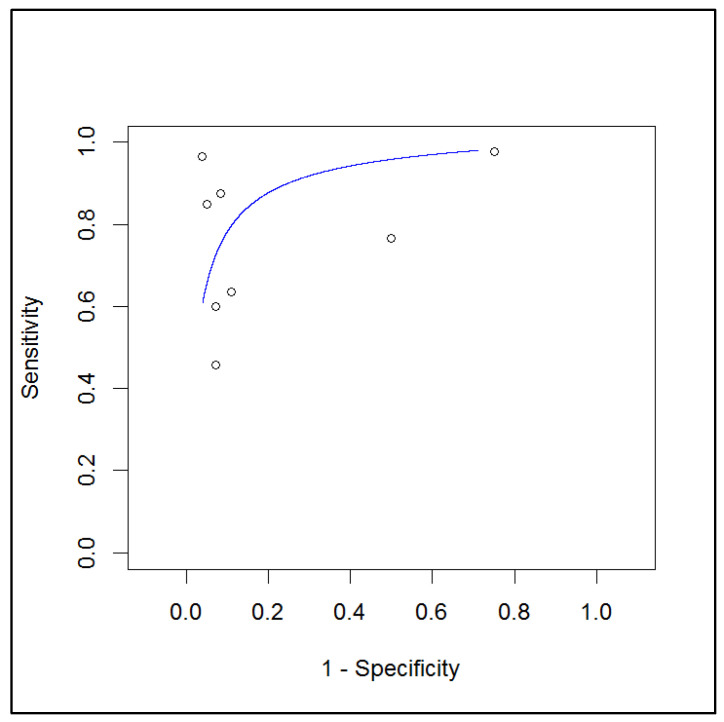
SROC curve of index test’s diagnostic accuracy in lymph node metastases.

**Figure 5 ijms-24-10136-f005:**
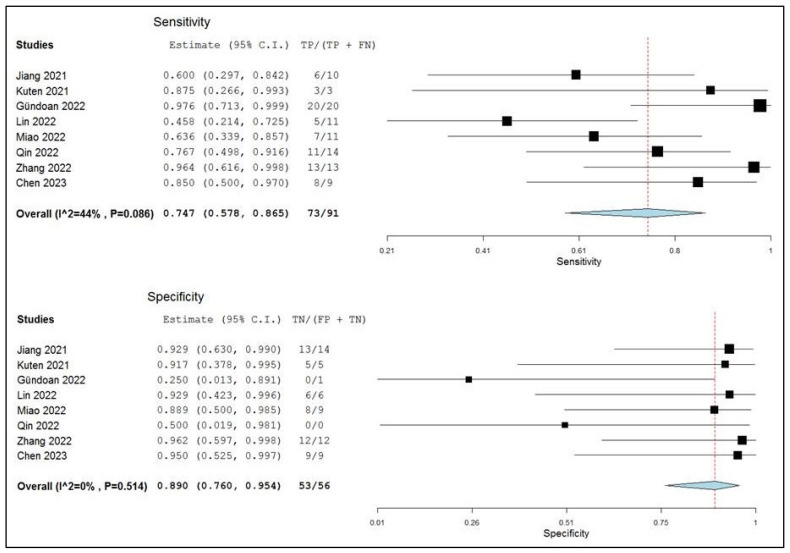
Sensitivity and specificity of the index test in the assessment of lymph node metastases and relative forest plots. Legend: 95% C.I.: 95% confidence interval; TP: true positive; TN: true negative; FP: false positive; FN: false negative, [14,15,17,18,19,20,21,22].

**Figure 6 ijms-24-10136-f006:**
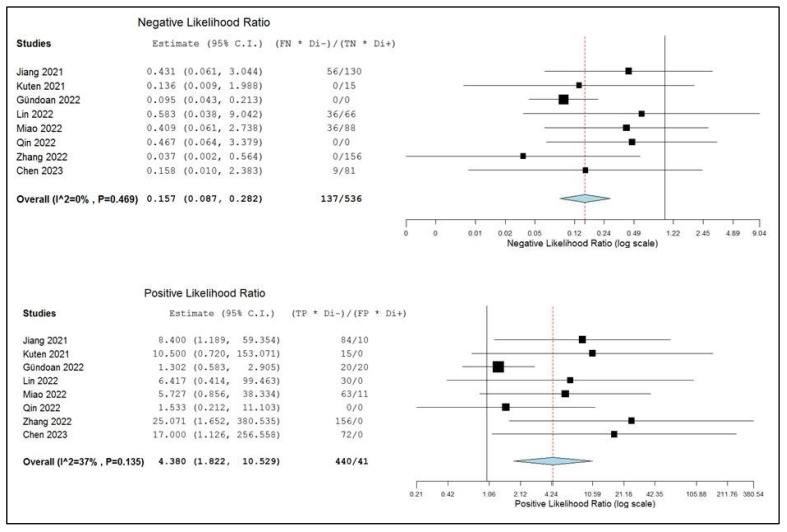
Negative and positive likelihood ratios of the index test in the assessment of lymph node metastases and relative forest plots. Legend: 95% C.I.: 95% confidence interval; TP: true positive; TN: true negative; FP: false positive; FN: false negative, [14,15,17,18,19,20,21,22].

**Figure 7 ijms-24-10136-f007:**
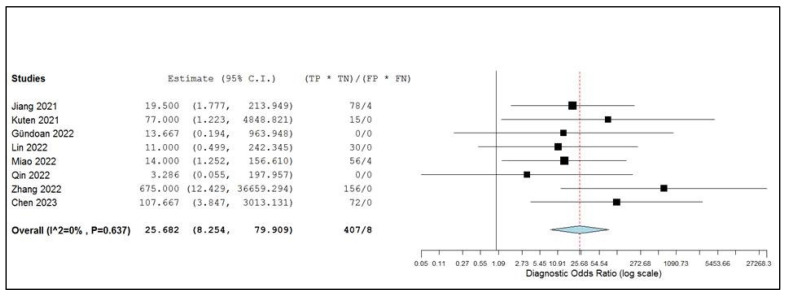
Diagnostic odds ratio of the index test in the assessment of lymph node metastases and relative forest plots. Legend: 95% C.I.: 95% confidence interval; TP: true positive; TN: true negative; FP: false positive; FN: false negative, [14,15,17,18,19,20,21,22].

**Figure 8 ijms-24-10136-f008:**
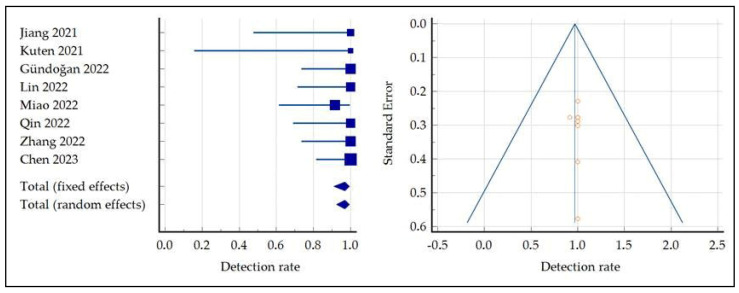
Meta-analysis and funnel plot concerning the detection rate of FAP-targeted PET in GC distant metastases, [14,15,17,18,19,20,21,22].

**Table 1 ijms-24-10136-t001:** General study information.

Authors [Ref.]	Year	Country	Study Design/Number of Involved Centres	Funding Sources
Jiang et al. [14]	2021	China	Retrospective/Bicentric	Startup Fund of Huashan Hospital, Fudan University; Shanghai Municipal Key Clinical Specialty; Shanghai Municipal Science and Technology Major Project; Shanghai Municipal Health Commission Fund
Kuten et al. [15]	2021	Israel	Prospective/Monocentric	None declared
Rong et al. [16]	2022	China	Retrospective/Monocentric	National Natural Science Foundation of China; China Postdoctoral Science Foundation; Science and Technology Planning Project of Guangzhou.
Gündoğan et al. [17]	2022	Turkey	Prospective/Monocentric	None declared
Lin et al. [18]	2022	China	Prospective/Monocentric	National Natural Science Foundation of China; Natural Science Foundation of Fujian Province; Fujian Provincial Health Technology Project; Startup Fund for Scientific Research of Fujian Medical University
Miao et al. [19]	2022	China	Prospective/Monocentric	Shanghai Municipal Key Clinical Specialty; Joint Research Development Project between Shenkang and United Imaging on Clinical Research and Translation
Qin et al. [20]	2022	China	Prospective/Monocentric	None declared
Zhang et al. [21]	2022	China	Retrospective/Monocentric	Research foundation projects from Luzhou Science and Technology Department; Affiliated Hospital of Southwest Medical University; Nuclear Medicine and Molecular Imaging Key Laboratory of Sichuan Province Open Project
Chen et al. [22]	2023	China	Retrospective/Multicentric	National Natural Science Foundation of China; Key Medical and Health Projects in Xiamen

**Table 2 ijms-24-10136-t002:** Patient key characteristics and clinical settings.

Authors [Ref.]	Sample Size (No. of Patients)	Mean/Median Age (Years)	Gender (Male %)	No. of Patients and Clinical Setting	GC Subtype (No. of Patients)	Comparative Imaging
Jiang et al. [14]	38	Mean: 63.7	76%	38 Staging	31 ADC 7 GSRCC	[^18^F]FDG PET/CT; [^18^F]FDG PET/MR
Kuten et al. [15]	13	Median: 70	46%	10 Staging 3 Restaging	9 ADC 4 GSRCC	[^18^F]FDG PET/CT
Rong et al. [16]	21	n.a.	n.a.	21 Restaging before immunotherapy	n.a.	[^18^F]FDG PET/CT
Gündoğan et al. [17]	21	Median: 61	57%	15 Staging 6 Restaging	17 ADC 3 GSRCC 1 mucinous carcinoma	[^18^F]FDG PET/CT
Lin et al. [18]	56	Median: 63.8	71%	45 Staging 11 Restaging	17 ADC 28 GSRCC	[^18^F]FDG PET/CT
Miao et al. [19]	62	Median: 64	71%	62 Staging	27 PCC 35 non-PCC	[^18^F]FDG PET/CT
Qin et al. [20]	20	Median: 56	45%	14 Staging 6 Restaging	9 ADC 4 ADC, partial SGRCC 4 GSRCC 2 PCC 1SCC	[^18^F]FDG PET/CT
Zhang et al. [21]	25	Mean: 56	48%	17 Staging 8 Restaging	18 ADC 6 SGRCC 1 GSRCC + mucinous carcinoma	[^18^F]FDG PET/CT
Chen et al. [22]	34	Median: 51	47%	22 Staging 12 Restaging	34 GSRCC	[^18^F]FDG PET/CT; [^18^F]FDG PET/MR

Legend: ADC: adenocarcinoma; CT: computed tomography; [^18^F]FDG: fluorodeoxyglucose; GSRCC: gastric signet ring cell carcinoma; MR: magnetic resonance; n.a.: not available; PCC: poorly cohesive carcinoma; PET: positron emission tomography.

**Table 3 ijms-24-10136-t003:** Index test key characteristics.

Authors [Ref.]	Tracer	Hybrid Imaging	Tomograph	Administered Activity	Uptake Time (Minutes)	Image Analysis
Jiang et al. [14]	[^68^Ga]Ga-DOTA-FAPi-04	PET/CT; PET/MR	PET/CT: Biograph mCT (Siemens^®^, Munich, Germany), Ingenuity TF (Philips^®^, Cambridge, MA, USA), uMI510 (United Imaging^®^, Shanghai, China); PET/MR: uPMR790 (United Imaging^®^, Shanghai, China)	111–185 MBq	60	Qualitative, semiquantitative (SUV_max_, TBR)
Kuten et al. [15]	[^68^Ga]Ga-DOTA-FAPi-04	PET/CT	NR	1.8–2.2 MBq/kg	60	Qualitative, semiquantitative (SUV_max_, TBR)
Rong et al. [16]	[^68^Ga]Ga-DOTA-FAPi-04	PET/CT	uEXPLORER (United Imaging ^®^, Shanghai, China)	1.8–2.2 MBq/kg	60	NR
Gündoğan et al. [17]	[^68^Ga]Ga-DOTA-FAPi-04	PET/CT	Discovery IQ (GE^®^, Boston, MA, USA)	2 MBq/kg	60	Qualitative, semiquantitative (SUV_max_, TBR)
Lin et al. [18]	[^68^Ga]Ga-DOTA-FAPi-04	PET/CT	Biograph mCT64 (Siemens^®^, Munich, Germany)	111–185 MBq	35–71	Qualitative, semiquantitative (SUV_max_, TBR)
Miao et al. [19]	[^68^Ga]Ga-DOTA-FAPi-04	PET/CT	Biograph Vision 450 (Siemens^®^, Munich, Germany)	1.85–2.96 MBq/kg	30–60	Qualitative, semiquantitative (SUV_max_, TBR)
Qin et al. [20]	[^68^Ga]Ga-DOTA-FAPi-04	PET/MR	SIGNA (GE^®^, Boston, MA, USA)	1.85–3.7 MBq/kg	30–60	Qualitative, semiquantitative (SUV_max_)
Zhang et al. [21]	[^68^Ga]Ga-DOTA-FAPi-04	PET/CT	uMI780 (United Imaging^®^, Shanghai, China)	1.85 MBq/Kg	60	Qualitrative, semiquantitative (SUV_max_)
Chen et al. [22]	[^68^Ga]Ga-DOTA-FAPi-04	PET/CT; PET/MR	PET/CT: Discovery MI (GE^®^, Boston, MA, USA), Biograph mCT (Siemens^®^, Munich, Germany); PET/MR: uPMR790 TOF (United Imaging^®^, Shanghai, China)	194.3 MBq	60	Qualitative, semiquantitative (SUV_max_, TBR)

Legend: CT: computed tomography, DOTA: 1,4,7,10-tetracetic-1,4,7,10-tetraazaciclododecan acid; FAPi: fibroblast-activation protein inhibitor, MR: magnetic resonance; NR: not reported; PET: positron emission tomography; TBR: target-to-background ratio.

**Table 4 ijms-24-10136-t004:** Outcomes of the included studies.

Authors [Ref.]	Aim of the Study	Primitive Lesion SUV_max_	Metastatic Lesions SUV_max_	Outcome
Jiang et al. [14]	Assess diagnostic accuracy of FAPi PET	Mean: 7.4 ± 5.0	n.a.	FAPi PET is superior to [^18^F]FDG PET for the detection of primary gastric cancers
Kuten et al. [15]	Assess diagnostic accuracy of FAPi PET	Median: 5.5	Lymph nodes: 4.3	FAPi PET is superior to [^18^F]FDG PET for the detection of primary gastric cancers and recurrences
Rong et al. [16]	Evaluate FAPi PET performance in predicting response to immunotherapy	n.a.	n.a.	High FAPi uptake is associated with a worse response to immunotherapy
Gündoğan et al. [17]	Assess diagnostic accuracy of FAPi PET	Median: 11.0	Lymph nodes: 5.7 Liver: 6.8 Bone: 4.8 Peritoneum: 5.7	FAPi PET could detect more lesions than [^18^F]FDG PET
Lin et al. [18]	Assess diagnostic accuracy of FAPi PET	Mean: 10.3	Lymph nodes: 6.3	FAPi PET is comparable to [^18^F]FDG PET in detecting primary tumors but outperformed [^18^F]FDG PET in detecting bone and peritoneal metastases
Miao et al. [19]	Assess diagnostic accuracy of FAPi and [^18^F]FDG dual-tracer PET/CT	Median: 18.81	n.a.	FAPi and [^18^F]FDG dual-tracer PET/CT were complementary and improved the sensitivity of detecting pre-treatment distant metastases
Qin et al. [20]	Comparison of diagnostic accuracy between FAPi PET/MR and [^18^F]FDG PET/CT	Mean: 11.31 ± 3.96	Lymph nodes: 6.58 ± 2.78 Peritoneum: 7.60 ± 5.85 Ovaries: 4.19 ± 1.72 Liver: 5.63 ± 1.96 Bone: 5.8 ± 5.39	Compared with [^18^F]FDG PET/CT, FAPi PET/MR had superior detection capabilities for primary tumors and metastases
Zhang et al. [21]	Comparison of diagnostic accuracy between FAPi PET and [^18^F]FDG PET	Median: 10.28	Lymph nodes: 9.2 Distant metastases: 8.0	FAPi PET is superior to [^18^F]FDG PET for the detection of primary tumor, lymph node, and distant metastases in patients with GC
Chen et al. [22]	Comparison of diagnostic accuracy between FAPi PET and [^18^F]FDG PET	Median: 5.2	Lymph nodes: 6.8 Bone and visceral metastases: 6.5 Uncommon sites: 6.0	FAPi PET had greater sensitivity and accuracy than [^18^F]FDG PET

Legend: CT: computed tomography; FAPi: fibroblast-activation protein inhibitor; [^18^F]FDG: fluorodeoxyglucose; MR: magnetic resonance; n.a.: not available: PET: positron emission tomography; SUV: standard uptake value.

**Table 5 ijms-24-10136-t005:** Meta-analysis of primary tumor detection rate.

Study	Sample Size	Detection Rate (%)	95% CI	Weight (%)	
				Fixed	Random
Jiang et al. [14]	38	100	90.7–100	16.7	14.5
Kuten et al. [15]	10	100	69.1–100	4.7	8.9
Gündoğan et al. [17]	15	100	78.2–100	6.9	10.7
Lin et al. [18]	45	100	92.1–100	19.7	15.1
Miao et al. [19]	62	90.3	80.1–96.3	27	16.1
Qin et al. [20]	14	100	76.8–100	6.4	10.4
Zhang et al. [21]	19	94.7	73.9–99.9	8.6	11.8
Chen et al. [22]	22	72.7	49.8–89.3	9.9	12.4
Total (fixed effects)	225	95.2	91.7–97.6	100	100
Total (random effects)	225	95.3	89–99	100	100

Legend: CI: confidence interval.

**Table 6 ijms-24-10136-t006:** Meta-analysis of distant metastases detection rate.

Study	Sample Size	Proportion (%)	95% CI	Weight (%)	
				Fixed	Random
Jiang et al. [14]	5	100	47.8–100	6.7	6.7
Kuten et al. [15]	2	100	15.8–100	3.3	3.3
Gündoğan et al. [17]	12	100	73.5–100	14.4	14.4
Lin et al. [18]	11	100	71.5–100	13.3	13.3
Miao et al. [19]	12	91.67	61.5–99.8	14.4	14.4
Qin et al. [20]	10	100	69.1–100	12.2	12.2
Zhang et al. [21]	12	100	73.5–100	14.4	14.4
Chen et al. [22]	18	100	81.5–100	21.1	21.1
Total (fixed effects)	82	96.9	90.9–99.4	100	100
Total (random effects)	82	96.9	92.3–99.4	100	100

Legend: CI: confidence interval.

## Data Availability

The data presented in this study are available upon request from the corresponding author.

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
