# Peer review of "Diagnostic Performance of Positron Emission Tomography with Fibroblast-Activating Protein Inhibitors in Gastric Cancer: A Systematic Review and Meta-Analysis"

_ijms, 2023, doi:10.3390/ijms241210136_

Round 1
Reviewer 1 Report
The main question addressed by the research is if FAP is a useful marker for PET staging in gastric cancer. The topic is relevant in the field. New clinical markers are useful for PET staging as the current FDG/PET lacks specificity. This paper adds further characterization of the clinical value of FAP compared with other published material. The conclusions are consistent with the evidence and arguments presented, and they address the main question posed. The references are appropriate and need extensive review and critical analysis of the literature.
Author Response
We thank the reviewer for having appreciated our extensive review.
Reviewer 2 Report
Page 17 mentions some limitations of the systematic review and meta-analysis, such as the lack of studies from other continents, but it does not specify which article it is referring to. Similarly, Page 1 provides information about a systematic review and meta-analysis on the diagnostic performance of Positron Emission Tomography with Fibroblast-Activating Protein Inhibitors in Gastric Cancer, but it does not mention any weaknesses of the article.
Good
Author Response
We thank the reviewer for the precious comments/suggestions.
We have modified the manuscript accordingly.
Reviewer 3 Report
These are some comments and points which should be under consideration:
The compilation of studies that involves large data sets is a difficult, if not impossible, task. However, when a study includes a quite low number of samples (e.g. 2, 5…), its particular value and its influence should be low. Thus it might be removed from further analysis.
In the case of SROC graph, the upper and lower limits on the axles should be 0-1.
The quality of the figures should be improved. The captions of the figures should be shorter. The explanation of each of the figures might be in the text.
The authors should enrich the text with their plans for the future work in this specific research area.
Author Response
Thanks for the comment, the reason why we added in the analysis also studies with low sample size is to guarantee transparency. Taking into account the low sample size, the statistical influence is low; therefore, these studies does not significant affect the pooled analysis.
All figures but figure 1 and 2 were captured from statistical softwares outputs (MedCalc and Meta[analyst], so they cannot be furtherly modified.
Concerning future research areas, hints and suggestions are available in the text in the discussion section. We have reported in the conclusions that more multicentric trials are needed to confirm the findings reported in this systematic review and meta-analysis.